# Biological Evaluation of Photodynamic Effect Mediated by Nanoparticles with Embedded Porphyrin Photosensitizer

**DOI:** 10.3390/ijms23073588

**Published:** 2022-03-25

**Authors:** Ludmila Žárská, Zuzana Malá, Kateřina Langová, Lukáš Malina, Svatopluk Binder, Robert Bajgar, Petr Henke, Jiří Mosinger, Hana Kolářová

**Affiliations:** 1Department of Medical Biophysics, Faculty of Medicine and Dentistry, Palacky University Olomouc, Hnevotinska 3, 775 15 Olomouc, Czech Republic; ludmila.zarska@centrum.cz (L.Ž.); zuzana.mala@upol.cz (Z.M.); katerina.langova@upol.cz (K.L.); lukas.malina@upol.cz (L.M.); svatopluk.binder@upol.cz (S.B.); robert.bajgar@upol.cz (R.B.); 2The Institute of Molecular and Translation Medicine, Faculty of Medicine and Dentistry, Palacky University Olomouc, Hnevotinska 5, 775 15 Olomouc, Czech Republic; 3Department of Inorganic Chemistry, Faculty of Science, Charles University, Hlavova 2030, 128 43 Prague, Czech Republic; henke@natur.cuni.cz (P.H.); mosinger@natur.cuni.cz (J.M.)

**Keywords:** photodynamic effect, cancer, nanoparticles

## Abstract

Clinically approved photodynamic therapy (PDT) is a minimally invasive treatment procedure that uses three key components: photosensitization, a light source, and tissue oxygen. However, the photodynamic effect is limited by both the photophysical properties of photosensitizers as well as their low selectivity, leading to damage to adjacent normal tissue and/or inadequate biodistribution. Nanoparticles (NPs) represent a new option for PDT that can overcome most of the limitations of conventional photosensitizers and can also promote photosensitizer accumulation in target cells through enhanced permeation and retention effects. In this in vitro study, the photodynamic effect of TPP photosensitizers embedded in polystyrene nanoparticles was observed on the non-tumor *NIH3T3* cell line and *HeLa* and *G361* tumor cell lines. The efficacy was evaluated by viability assay, while reactive oxygen species production, changes in membrane mitochondrial potential, and morphological changes before and after treatment were imaged by atomic force microscopy. The tested nanoparticles with embedded TPP were found to become cytotoxic only after activation by blue light (414 nm) due to the production of reactive oxygen species. The photodynamic effect observed in this evaluation was significantly higher in both tumor lines than the effect observed in the non-tumor line, and the resulting phototoxicity depended on the concentration of photosensitizer and irradiation time.

## 1. Introduction

Currently, photodynamic therapy is an alternative treatment for various diseases, including cancer (esophagus, bladder, cervix, etc.), and especially for the treatment of superficial tumors (melanoma) [1,2,3]. The applied photosensitizer is selectively absorbed by the tumor cells. Subsequently, the tumor is irradiated with visible light to absorption bands of photosensitizer (PS). The phototoxicity is initiated by the absorption of photons by PS, followed by energy and/or electron transfer to mainly triplet oxygen that leads to the generation of singlet oxygen, O_2_(^1^Δ_g_) and/or other reactive oxygen species (ROS) and subsequently to the oxidation and degradation of vital biomolecules. Especially short-living O_2_(^1^Δ_g_) (ca. 3.5 µs in aqueous media) plays a key role in spreading the initial damage to biomolecules that leads to vascular collapse, tissue destruction, and cell death. Thus, PDT induces necrosis and/or apoptosis of tumor cells (or other target cells) by producing ROS with photoactivated PS [4,5,6,7]. However, PDT is limited by the (photo)properties of PS, especially low quantum yields of ROS in the target environment. An important limiting factor is also a low selectivity of PS with associated photodynamic damage of adjacent normal tissues and secondary damage due to the inflammatory response. Likewise, insufficient biological distribution and pharmacokinetics are the most common limiting factors of PSs [8].

The nanotechnological approach is one of the options to overcome and complement these limitations [9,10]. Nanotechnology is related to the understanding and control of particles with dimensions ranging from 1 to 100 nm, where unique physical and chemical properties are generated that can be used in diagnostic and therapeutic practice [11,12]. A major advantage of NP is the large surface-to-volume ratio [13] that can effectively increase the amount of PS delivered [14]. Nanocarriers are also attributed to the amphiphilic nature of PS, which allows nanoparticles with bound PS to travel indefinitely through the bloodstream without inactivation by plasma components [15,16]. The fact that nanoparticles are on the same length scale as many biological structures (e.g., proteins or viruses) supports their privileged entry into biological systems [17]. The proportion of NP drug accumulation in the tumor is significantly higher than in normal tissue due to leakage from the endothelium of the tumor vascular system. This phenomenon is known as the enhanced permeability effect (EPR). The defect in the lymphatic system (drainage system) leads to the retention of nanoparticles in the tumor. This retention is known as increased retention. Both phenomena are collectively referred to as the EPR effect [18]. By exploiting this EPR effect, the concentration of anticancer drugs in the tumor is increased many folds, compared with healthy body tissue. In normal tissues, extracellular fluid is constantly discharged into the lymphatic vessels. This allows a constant outflow and recovery of interstitial fluid, as well as the return of extravasant solutions and colloids back into the circulation [19,20]. Polystyrene nanoparticles are widely used for cellular uptake [21], in vitro [22], and in vivo [23] assays. As shown in previous studies, the cytotoxicity of this material depends on the particle size [24], the functional groups [25] used, and, last but not least, the type of cell line tested [22]. Polystyrene particles interfere with cellular metabolism mainly by binding to macromolecules and by the formation of reactive oxygen species [22]. In this in vitro study, aqueous dispersion of highly sulfonated polystyrene NP with embedded 5,10,15,20-tetraphenylporphyrin (TPP) (Figure 1) was tested. The photodynamic effect of TPP-NP in the presence of bacterial strains was already observed in our previous study [26]; based on these results, the testing of TPP embedded NP was extended to the non-tumor *NIH3T3* cell line and tumor cell lines *HeLa* and *G361*. Cell viability, ROS generation, and changes in mitochondrial membrane potential were studied. Morphological changes in cell culture before and after PDT were displayed by atomic force microscopy (AFM).

## 2. Results

### 2.1. Measurement of Dark Toxicity

The dark toxicity of TPP-NPs (Figure 2) was assessed on *HeLa*, *G361*, and *NIH3T3* cells. Used concentrations of samples were 0.05, 0.09, 0.19, 0.38, 0.75, and 1.5 × 10^13^ NP/mL. No significant changes in viability were observed in the tested TPP-NP. Measured viability values after TPP-NP application ranged from 90% to 105% in all used cell cultures.

### 2.2. Cancer Cell Cytotoxicity—MTT Assay

Cell viability after treatment with TPP-NP was determined by MTT assay. Significant changes were observed at each combination of concentration and irradiation time used. In the non-tumor line (Figure 3), we observed a gradual decrease in viability with increasing concentration. Viability after 5 and 10 min of irradiation at the same concentration was comparable. In *HeLa* cells (Figure 4), after irradiation with irradiation times of 0.5 and 1 min at concentrations of 0.05–0.75 × 10^13^ NP/mL, a decrease in cell viability with increasing concentration was observed. In contrast, this significant trend was not observed at irradiation times of 5 and 10 min. Cytotoxicity in *G361* cells (Figure 5) increased with increasing concentration of TPP-NP after 0.5 min of irradiation, and the same trend was observed after 1 min of irradiation for concentrations in the range of 0.05–0.38 × 10^13^ NP/mL. After 5 and 10 min of irradiation, this trend was not observed, similarly to *HeLa* cells. The least sensitive cell line to TPP-NPwas non-tumor *NIH3T3*, compared with both tumor lines tested.

### 2.3. Measurement of ROS Production

ROS production of TPP-NPs was tested using a CM-H2DCFDA fluorescence probe (general oxidative stress indicator). Significant changes in ROS production after irradiation were observed in all cell cultures. The trend of increasing ROS fluorescence with increasing concentration and irradiation time was observed for all tested combinations of concentration and irradiation only for the non-tumor line *NIH3T3*. For the *HeLa* tumor line, this trend was observed after irradiation at 0.5 and 1 min at concentrations in ranges of 0.05–0.75 × 10^13^ NP/mL and 0.05–0.38 × 10^13^ NP/mL, respectively. In *G361* cells, after irradiation for 0.5 min, we observed increasing ROS production at all tested concentrations; after irradiation for 1 min, the trend was observed in the range of 0.05–0.38 × 10^13^ NP/mL. In both tumor lines, we no longer observed this trend after 5 and 10 min irradiation times. The highest ROS production was observed in the *HeLa* tumor line after 10 min of irradiation. The ROS production levels induced by TPP-NPs in *NIH3T3* (Figure 6), *HeLa* (Figure 7), and *G361* (Figure 8) cells are shown.

### 2.4. Mitochondrial Membrane Potential (MMP)

Mitochondrial membrane potential was measured using a JC-1 fluorescence probe. Results are expressed as green/red fluorescence ratio. The higher the JC-1 fluorescence ratio, the higher the cell damage. The results showed increasing values of the green/red fluorescence ratio depending on increasing concentrations of TPP-NP at all irradiation times and tested concentrations in the non-tumor line *NIH3T3* (Figure 9).

This trend was observed in *HeLa* cells (Figure 10) only after 0.5 and 1 min at concentrations of 0.05–0.38 × 10^13^ NP/mL and 0.05–0.75 × 10^13^ NP/mL, respectively. For *G361* (Figure 11), after irradiation for 0.5 min, the value of the green/red fluorescence ratio increased at all tested concentrations, while at 1 min, this was in the range of 0.05–0.75 × 10^13^ NP/mL. For irradiation times of 5 and 10 min, no substantial changes in the green/red fluorescence ratio with respect to increasing concentration were observed in any of the tumor lines.

### 2.5. Atomic Force Microscope Image

Nanoparticles with embedded TPP were imaged in 2D (Figure 12A) and 3D (Figure 12B) by atomic force microscopy. Based on the above test results (MTT, ROS, MMP), topographic changes before and after PDT therapy were visualized in *HeLa* and *G361* cancer cell cultures. Figure 12C shows *HeLa* cells, and Figure 12E shows *G361* cells, before irradiation. In both cases, we observed intact cells of typically elongated shape. *HeLa* cells after irradiation with an irradiation time of 1 min and using a TPP-NP concentration of 0.09 × 10^13^ NP/mL are shown in Figure 12D. *G361* cells are shown in Figure 12F, with a TPP-NP concentration of 0.09 × 10^13^ NP/mL and irradiation with an irradiation time of 1 min. In both, we observed that the cell cultures were disrupted, and the cells lost their original elongated shape and were destroyed.

## 3. Discussion

The delivery of hydrophobic porphyrins to the target sites is one of the main challenges in PDT to be overcome. NPs are able to spontaneously accumulate in solid tumors through the EPR effect due to a combination of leaky vasculature, poor lymphatic drainage, and increased vessel permeability [27,28]. Encapsulating or attaching photosensitizers to NPs makes them a more suitable strategy for tissue delivery because many NP features such as specific targeting, the kinetics of uptake, immune tolerance, NP charge, as well as other characteristics for NPs can be created through rational design [29]. In our study, we focused on biological *in vitro* testing of aqueous dispersions of stable highly sulfonated polystyrene NPs with encapsulated hydrophobic TPP photosensitizer. The encapsulated TPP was well protected against external quenchers and aggregation by the shell of the polystyrene with high oxygen permeability, which enabled quenching of triplet states of TPP exclusively by oxygen and transport of O_2_(^1^Δ_g_) and/or other ROS to biological targets outside of the NPs.

TPP-NPs were found to induce more significant cytotoxic effects in both *HeLa* and *G361* tumor cells at all applied concentrations and irradiation times, compared with the non-tumor *NIH3T3* cell line. In contrast, the application of TPP-NPs without access to visible light had a negligible effect on the survival of the tested cells. 

In this study, this trend was observed in the non-tumor line *NIH3T3* at all observed concentrations and irradiation times, in contrast to the *HeLa* and *G361* tumor cell lines, in which the trend appeared only after irradiation times of 0.5 and 1 min at concentrations in ranges of 0.05–0.75 × 1013 NP/mL and 0.05–0.75 × 10^13^ NP/mL, respectively. For irradiation times longer than 1 min, ROS production was not dependent on the concentration. The data obtained show that higher photosensitizer concentrations increased ROS production and decreased cell viability. However, this trend was no longer observed in tumor cell lines tested at higher irradiation times, whatever concentration of TPP-NPs was used. A similar result indicating concentration independence was published in a previous study [30], in which the viability of *NIH3T3* cells after application of water-soluble porphyrin photosensitizers and irradiation reached the same value at all concentrations tested.

By comparing the effect of TPP-NPs on *HeLa* and *G361* cell viability, a greater cytotoxic effect was observed for *HeLa*. These results were supported by the ROS assay, which monitors ROS production in cells. The dependence of ROS production on TPP-NP concentration and irradiation times of 0.5 and 1 min was only observed in both cancer cell lines. In contrast, in *NIH3T3* cells, this concentration-dependent ROS production was observed at all tested irradiation times. 

Higher photogeneration of ROS after TPP-NP activation was observed in *Hela* cells. The same fact, i.e., the increased sensitivity of *HeLa* cells to oxidative stress, was also observed in previous studies [31,32,33]. A possible explanation is the reduced amount of glutathione in the *HeLa* and at the same time constant level of glutathione during hyperoxia, which causes increased sensitivity in *HeLa* cells, thereby destroying ROS [32].

Generally, high amounts of ROS can increase the permeability of lysosomal membranes, which leads to the release of lysosomal proteases, resulting in changes in mitochondrial membrane potential and cell death [34,35]. This trend was monitored by the MMP assay, which showed greater changes in membrane potential in *HeLa* and *G361* tumor cells than that in *NIH3T3*. The obtained MMP assay data were in accordance with the results of the MTT assay and ROS production.

Cells were imaged by AFM before and after photodynamic treatment in order to observe changes on the cell surface after PDT induction. In general, the shape of the cells depends on the type and also on the state of the cell. Living undamaged cells have a typical elongate shape in comparison to photodynamically damaged cells [36,37]. Before photodynamic treatment, it was almost demanding to find any dead cell. 

In conclusion, PDT mediated by TPP-NPs had minimum phototoxic effects on the non-tumor *NIH3T3* cell line, in contrast to strong effects on tumor *HeLa* and *G361* cells.

## 4. Materials and Methods

### 4.1. Cell Lines

*HeLa* (cervical cancer) and *G361* (human skin malignant melanoma) tumor cell lines, as well as the non-tumor *NIH3T3* (embryonic mouse fibroblasts) cell line, were used for the MTT viability/phototoxicity test, ROS measurement, mitochondrial membrane potential measurement, and AFM imaging. *HeLa* and *G361* cell lines were taken from ATTC: The Global Bioresource Center, and NIH3T3 cells were cultured in Dulbecco’s modified Eagle medium (DMEM, Sigma Aldrich, MA, USA) in a thermobox at 37 °C and 5% CO_2_.

### 4.2. Tested Samples

Photodynamic activity of negatively charged singlet oxygen-generating NP (average diameter 15 ± 7 nm) with embedded hydrophobic TPP (5,10,15,20-tetraphenylporphyrin, Sigma Aldrich, MA, USA) (TPP-NP) was studied. TPP is one of the simplest porphyrin chromophores with a high quantum yield of singlet oxygen (Φ_Δ_ = 0.74 in CCl_4_) [38], the binding of which has no effect on the morphology or size of NP. TPP-NP with embedded TPP (3 mg/mL) were prepared by a modified top-down method of nanoprecipitation using sulfonated electrospun polystyrene nanofiber membranes with TPP, as published earlier [39,40]. Briefly, the fixed polystyrene membranes (250 cm^2^, typically 150 mg) on quartz substrates were immersed in 96% sulfuric acid at room temperature for 54 h. The treated membranes were washed repeatedly with deionized water until a neutral pH was reached. The wet membranes were then immersed in 16 mL of dry THF and stirred for 60 s. Embedded TPP in NPs (TPP-NPs) were prepared by enriching dry THF with 14 mg TPP, with a final yield of 10 wt%. TPP in NP. Subsequently, deionized water (80 mL) was added, and the THF was removed by evaporation in vacuo. Larger microparticles were separated from the NP dispersion by centrifugation (10 min at 4700 g). Traces of sulfuric acid and THF were removed from the NP dispersion by dialysis using a Float-A-Lyzer G2 membrane with a molecular weight cut-off of 50 kDa for 18 h in deionized water at room temperature. The stock dispersion with a TPP-NP concentration of ~3 × 10^13^ NPs/mL (with an embedded TPP concentration of 10% (*w*/*w*) in NPs) was stored in deionized water in the dark.

TPP-NP was also characterized by UV–Vis absorption spectroscopy (Appendix A). The Soret band at 421 nm and four Q bands were characteristic for the sample of TPP-NP with 10 wt % TPP (in nonpolar solvents such as toluene). This indicates that nonpolar TPP molecules are mainly in nonpolar polystyrene matrices in their monomeric form. Broadening Soret band or hypochromicity was not detected in the sample, indicating that there is no extensive aggregation of TPP. Additionally, emission fluorescence spectroscopy of the TPP-NP revealed that most of the porphyrin molecules were present in the monomeric state, well embedded in the polystyrene bulk; however, they were affected to some extent by protonation. Two bands with wavelengths 654–680 nm and 716–718 nm are typical for TPP in a hydrophobic environment (Appendix A).

Further photophysical and photochemical characterization of the tested TPP-NP has been described in detail in previous studies [39,41].

### 4.3. Light Source

A homemade LED-based light source containing 350 pieces of 5 mm LEDs, specifically designed for the irradiation of experimental microplates, was used. The light source is protected by a National Patent CZ 302,829 B6. 

The cells in 96-wells (P-Lab, Praha, Czech Republic) were irradiated with irradiation times of 0.5 min (light dose 1.6 J/cm^2^), 1 min (3.2 J/cm^2^), 5 min (16.2 J/cm^2^), and 10 min (32.4 J/cm^2^) by an LED-based light source with wavelength 414 nm and light irradiance 54 mW/cm^2^ at room temperature. Irradiation was determined using an ILT 1700 radiometer sensor SED033 (International Light Technology).

### 4.4. Cell Cytotoxicity (MTT) Assay

Cells loaded with TPP-NPs at concentrations of 0.05, 0.09, 0.19, 0.38, 0.75 and 1.5 × 10^13^ NP/mL were incubated in thermobox at 37 °C and 5% CO_2_. After 24 h incubation, DMEM was replaced by a phosphate-buffered saline solution (PBS, pH 7.4 own preparation), and the cells were then irradiated with irradiation times of 0.5, 1, 5, and 10 min by an LED-based light source with wavelength 414 nm at room temperature. After irradiation, fresh DMEM was added to the cell culture, and the 96-well plates were incubated for another 24 h in thermobox. Then, DMEM was replaced with PBS, followed by the addition of 50 μL of 0.5 mg/mL tetrazolium salt MTT (3- (4,5-dimethyl-2-thiazolyl) -2,5-diphenyl-2H-tetrazolium bromide, Sigma Aldrich) (dissolved in PBS), detecting the activity of dehydrogenases (enzymes), which reflect the number of viable cells present. The tetrazolium group was incorporated into metabolically active mitochondria for 4 h of incubation at 37 °C and 5% CO_2_ (this reaction does not occur in dead cells). After carefully replacing the MTT solution, 100 μL DMSO (dimethyl sulfoxide, Sigma Aldrich, MA, USA) was used to dissolve the violet formazan crystals. The absorbance of the solution was evaluated by measuring in a 96-wells microplate reader (SYNERGY^TM^ HT, BioTek Instruments, VT, USA) at 570 nm. Three 96-well plates were used as the negative controls (cells inoculated with TPP-NPs without the irradiation). The measured data were calculated by Phototox Version 2.0 software (ZEBET, Germany). Dark phototoxicity was measured in parallel under identical conditions but without the irradiation of cell samples. The MTT assay procedure has been standardized and previously published by Zarska et al. (2021) [31].

### 4.5. ROS Production Measurement

The studied cells were incubated with TPP-NPs at 37 °C and 5% CO_2_ in a thermobox. After incubation (24 h), DMEM was replaced with a solution of 10 μM fluorescent probe CM-H_2_DCFDA (5-(α-6)-chloromethyl-20.70 dichlorodihydrofluorescein diacetate, Invitrogen) (Ex/Em: 495/530 nm). During a 30 min incubation inside *NIH3T3*, *HeLa*, and *G361* cells, CM-H_2_DCFDA is reduced to colorless CM-H_2_DCF by intracellular esterases and thiols. Due to the presence of ROS (formed after irradiation with an LED light source with a wavelength of 414 nm), it is oxidized to a DCF form (green fluorescent chloroform). ROS generation was measured with an SYNERGYTM HT microplate reader immediately after irradiation with irradiation times of 0.5, 1, 5, and 10 min, respectively. The value of the relative fluorescence unit is directly proportional to the H_2_O_2_ concentration; therefore, it directly reflects the increase and change in the amount of ROS present. The ROS production measurement procedure has been standardized and previously published by Zarska et al. (2021) [31].

### 4.6. Mitochondrial Membrane Potential (MMP) Assay

The MMP was monitored by a fluorescent cationic voltage-dependent dye JC-1 (5,5,6,6-tetrachloro-1,1′,3,3′ tetraethylbenzimidazolylcarbocyanine iodide, Sigma Aldrich). After 24 h incubation, *NIH3T3*, *HeLa*, and *G361* cells with TPP-NPs in thermobox (37 °C and 5% CO_2_) located in the 96-wells microplate were irradiated (0.5, 1, 5, and 10 min) by an LED-based light source with wavelength 414 nm. Immediately after irradiation, the cells were incubated with JC-1 at a final assay concentration of 2 μg/mL for 20 min at 37 °C, with 5% CO_2_, and then washed with PBS. In healthy cells, JC-1 enters activated mitochondria and forms aggregates that alter the fluorescent property of the JC-1 dye. Unhealthy or apoptotic cells have a low mitochondrial membrane potential. JC-1 does not form aggregates in mitochondria with low membrane potential and remains in monomeric form and shows green fluorescence, in contrast to aggregates producing intense red fluorescence. Fluorescent measurements were performed using microplate reader Synergy HT. The results were expressed as the ratio of the green fluorescence (excitation wavelength, 485 nm; emission wavelength, 548 nm) and red fluorescence (excitation wavelength, 520 nm; emission wavelength, 590 nm) retained within the cells. Thus, the higher the ratio of green to red fluorescence, the higher the polarization of the mitochondrial membrane. The MMP assay procedure has been standardized and previously published by Zarska et al. (2021) [31].

### 4.7. AFM Sample Preparation

*HeLa* and *G361* cells (2 × 10^5^) were seeded in 35 mm diameter Petri dishes (Willco, Amsterdam, Netherlands) containing 2 mL of culture medium (DMEM, Sigma Aldrich, MA, USA). Cancer cells were incubated (37 °C and 5% CO_2_) with TPP-NPs at a concentration 0.09 × 10^13^ NP/mL for 24 h. The cells were then washed in PBS and irradiated in 2 mL PBS with an irradiation time of 1 min. PBS was replaced with 2 mL of fresh DMEM, and the sample cells were kept in an incubator at 37 °C and 5% CO_2_ for 24 h. AFM measurements were then performed.

### 4.8. AFM Imaging

An Atomic Force Microscope Bioscope Catalyst (Bruker, Karlsruhe, Germany) was used to image the surface topography of sulfonated polystyrene-NP-embedded hydrophobic TPP and changes in the surface topography of photodynamically treated cancer cell lines. Cells were imaged at a scan rate of 0.1 Hz with 100 μm scan size. We used a ScanAssyst-FLUID + DNP-10-B tip and a spring constant of 0.12 Nm^−1^ on the nitride lever. AFM surface images were obtained in contactless mode.

### 4.9. Statistical Data Processing

Each sample was tested nine times; therefore, values represent mean ± standard deviation (SD) from 12 repetitions. To find the difference in efficacy of the photosensitizers, the *t*-test was performed at a statistical significance level of 0.05. Statistics SPSS Statistics for Windows Version 23.0 was used for statistical processing (Armonk, NY, USA: IBM Corp). The level of significance of the test is denoted by asterisks: * *p* < 0.05, ** *p* < 0.01, *** *p* < 0.001.

## 5. Conclusions

The results obtained in this *in vitro* study show that TPP-NP did not cause dark cytotoxicity in any of the tested cell lines. The tested TPP-encapsulated sulfonated polystyrene NP was cytotoxic in *HeLa* and *G361* cancer cells under PDT (blue light activation) conditions. When testing *HeLa* and *G361* tumor lines, we found almost identical efficacy of TPP-NP, which was independent in concentration and irradiation time (valid for 5 and 10 min irradiation) and also caused the greatest cytotoxicity. The observed phototoxic effect of PDT-mediated TPP-NP was lowest in the non-tumor *NIH3T3* cell line. The most pronounced cytotoxicity in *NIH3T3* was observed at a combined TPP-NP concentration of 1.5 × 10^13^ NP/mL and an irradiation time of 10 min. AFM measurement also showed substantial morphological and shape changes associated with cell death after PDT. It can be assumed that photosensitive TPP-NP can find successful applications in photodynamic therapy.

## Figures and Tables

**Figure 1 ijms-23-03588-f001:**
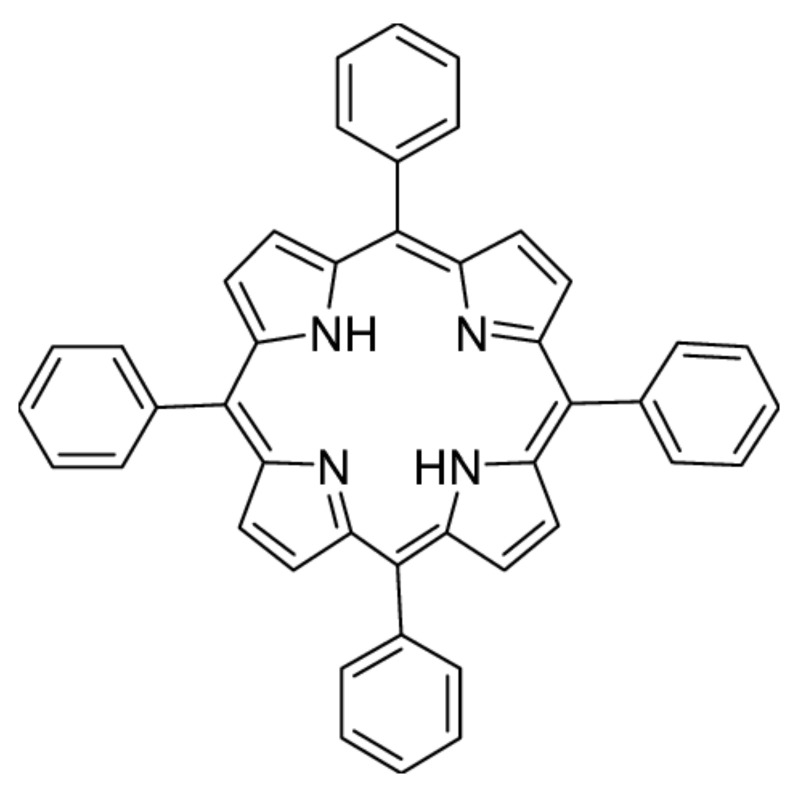
Chemical structure of 5,10,15,20-tetraphenylporphyrin (TPP).

**Figure 2 ijms-23-03588-f002:**
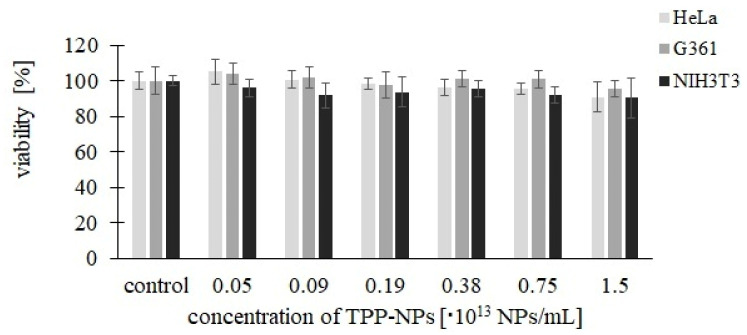
Measurement of TPP-NP dark toxicity. Each column in the graph represents the mean ± SD value calculated from twelve repetitions (triplets of four independent measurements) treated with the same TPP-NP concentration and the irradiation time. The control represents cells without the application of TPP-NPs.

**Figure 3 ijms-23-03588-f003:**
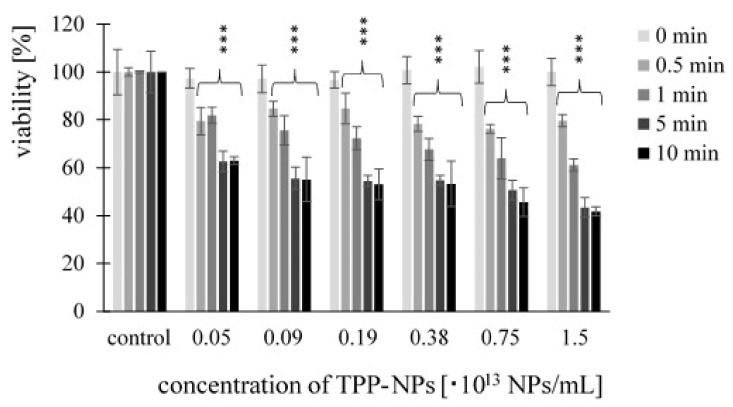
The dependence of *NIH3T3* cell viability on TPP-NP concentration. The dependence of cell viability on the concentration of TPP-NP was determined by MTT assay. Each column in the graph represents the mean ± SD value calculated from twelve repetitions (triplets of four independent measurements) treated with the same TPP-NP concentration and irradiation time. The control represents cells irradiated without TPP-NP (negative control), and its value was set as 100%. TPP-NP were compared with control; statistically significant results are marked with an asterisk, and the level of significance of the test is denoted as, *** *p* < 0.001.

**Figure 4 ijms-23-03588-f004:**
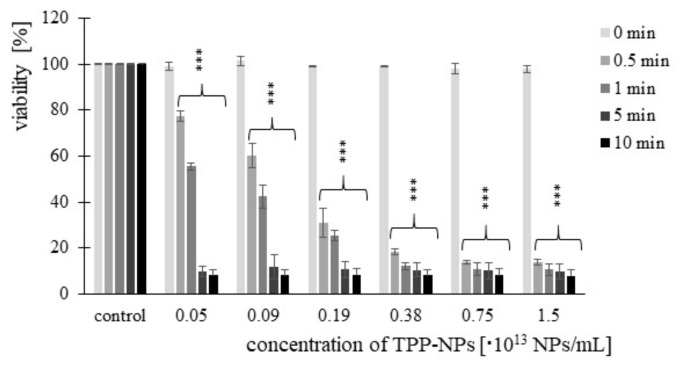
The viability of the cancer cell line *HeLa* treated with six different concentrations of TPP-NP and irradiated with irradiation times of 0.5, 1, 5, and 10 min. Each column in the graph represents the mean ± SD value calculated from twelve repetitions (triplets of four independent measurements) treated with the same TPP-NP concentration and the irradiation time. The control represents negative control (irradiated cells without TPP-NP), and its value was set as 100%. TPP-NP were compared with control; statistically significant results are marked with an asterisk, and the level of significance of the test is denoted as, *** *p* < 0.001.

**Figure 5 ijms-23-03588-f005:**
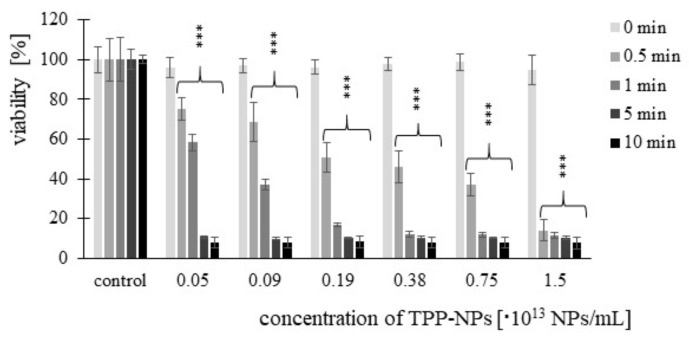
*G361* cancer cell line was evaluated for viability. The cells were treated with six different concentrations of TPP-NP and irradiated with irradiation times of 0.5, 1, 5, and 10 min. Each column in the graph represents the mean ± SD value calculated from twelve repetitions (triplets of four independent measurements) treated with the same TPP-NP concentration and the irradiation time. The control represents negative control (irradiated cells without TPP-NP), and its value was set as 100%. TPP-NP were compared with control; statistically significant results are marked with an asterisk, and the level of significance of the test is denoted as, *** *p* < 0.001.

**Figure 6 ijms-23-03588-f006:**
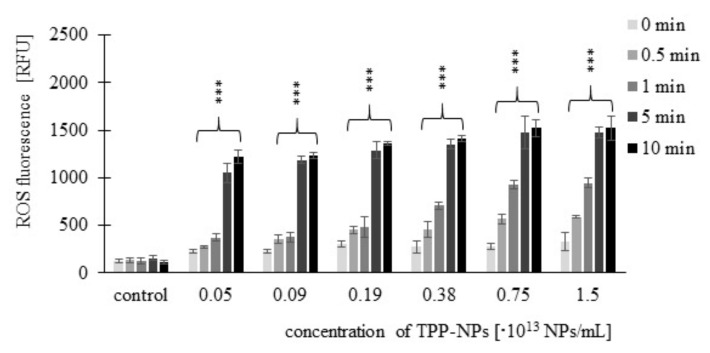
The dependence of ROS production in *NIH3T3* cells on the concentration of TPP-NP. ROS measurement was performed immediately after irradiation. The control represents cells irradiated without TPP-NP (the negative control). Data are presented as the mean ± SD of twelve repetitions (triplets of four independent measurements). TPP-NP were compared with control; statistically significant results are marked with an asterisk, and the level of significance of the test is denoted as, *** *p* < 0.001.

**Figure 7 ijms-23-03588-f007:**
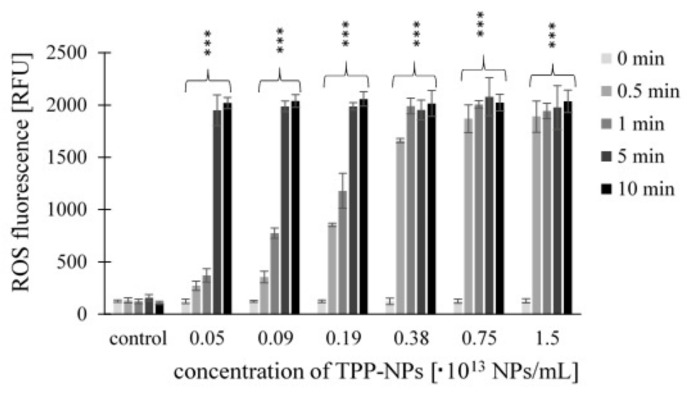
The fluorescence levels of ROS of the cancer cell line *HeLa* were measured. Cells were treated with highly sulfonated polystyrene NP with embedded TPP. ROS measurement was performed immediately after irradiation. The control represents negative control (irradiated cells without TPP-NP), and its value was set as 100%. Data are presented as the mean ± SD of twelve repetitions (triplets of four independent measurements). TPP-NP were compared with control; statistically significant results are marked with an asterisk, and the level of significance of the test is denoted as, *** *p* < 0.001.

**Figure 8 ijms-23-03588-f008:**
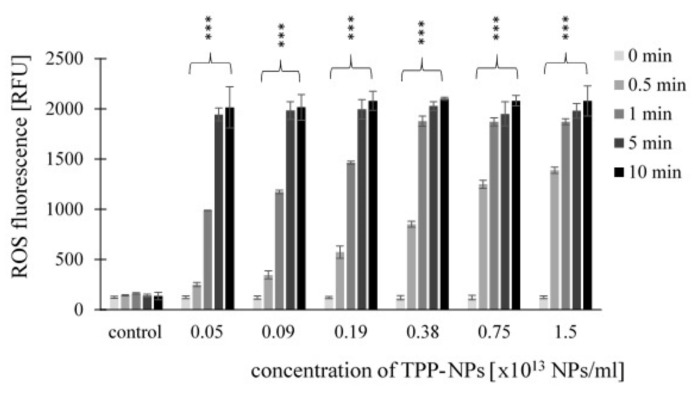
The ROS fluorescence of the *G361* cells treated with TPP-NP and irradiated with irradiation times of 0.5, 1, 5, and 10 min. ROS measurement was performed immediately after irradiation. The control represents cells irradiated without TPP-NP (the negative control). Data are presented as the mean ± SD of twelve repetitions (triplets of four independent measurements). TPP-NP were compared with control; statistically significant results are marked with an asterisk, and the level of significance of the test is denoted as, *** *p* < 0.001.

**Figure 9 ijms-23-03588-f009:**
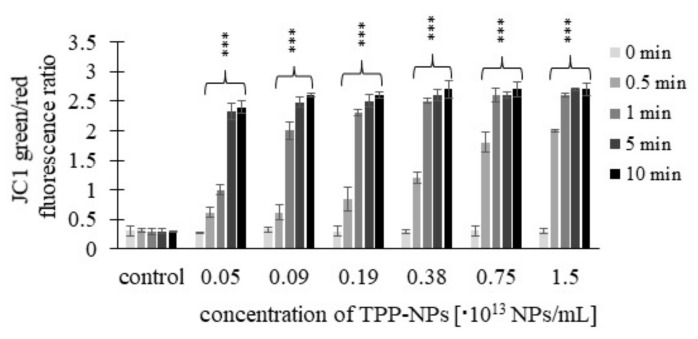
Effect of TPP-NPs on membrane potential in *NIH3T3* cells. The mitochondrial membrane potential of non-cancer cell line was measured after incubation with six different concentrations of TPP-NP. The higher the JC-1 green/red fluorescence ratio, the greater the cell damage. The control represents irradiated cells in the absence of TPP-NPs (negative control). Data are presented as the mean ± SD of twelve repetitions (triplets of four independent measurements). TPP-NP were compared to control; statistically significant results are marked with an asterisk, and the level of significance of the test is denoted as, *** *p* < 0.001.

**Figure 10 ijms-23-03588-f010:**
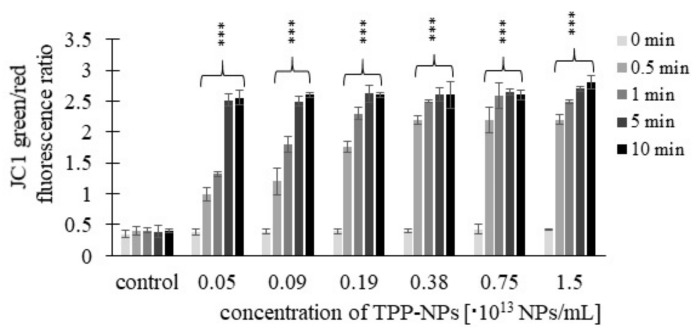
The fluorescence green/red ratio of *HeLa* cancer cell lines was measured after incubation with different concentrations of TPP-NP and irradiation with irradiation times of 0.5, 1, 5, and 10 min. The higher the JC-1 green/red fluorescence ratio, the greater the cell damage. The control represents irradiated cells in the absence of TPP-NPs (negative control). Data are presented as the mean ± SD of twelve repetitions (triplets of four independent measurements). TPP-NP were compared with control; statistically significant results are marked with an asterisk, and the level of significance of the test is denoted as, *** *p* < 0.001.

**Figure 11 ijms-23-03588-f011:**
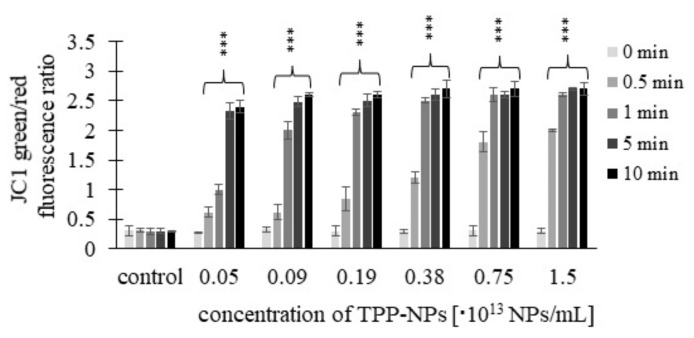
Effect of TPP-NP on membrane potential in *G361* cells. The mitochondrial membrane potential of cancer cell lines was measured after incubation with twelve different concentrations of TPP-NP. The higher the JC-1 green/red fluorescence ratio, the greater the cell damage. The control represents irradiated cells in the absence of TPP-NP (negative control). Data are presented as the mean ± SD of twelve repetitions (triplets of four independent measurements). TPP-NP were compared with control; statistically significant results are marked with an asterisk, and the level of significance of the test is denoted as, *** *p* < 0.001.

**Figure 12 ijms-23-03588-f012:**
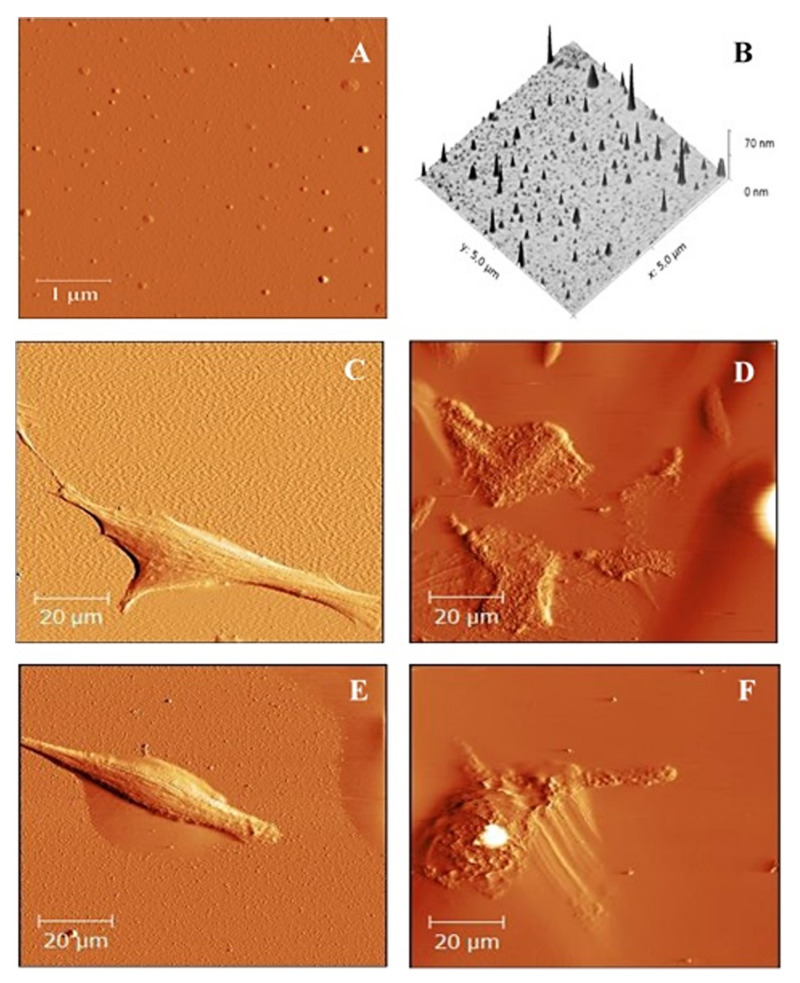
AFM 2D topography (**A**) and the corresponding 3D reconstructions (**B)** images of TPP-NPs. Scan area: 5 μm × 5 μm. *HeLa* (**C**) and G361 (**E**) cells were displayed before therapy. Images of *HeLa* cells after treatment by TPP-NPs (**D**) were taken with concentration 0.09 × 10^13^ NP/mL and irradiation time of 1 min. G361 cell line after treatment by TPP-NPs (**F**) with concentration 0.09 × 10^13^ NP/mL and irradiation time of 1 min are displayed. Images (**C**–**F**) were taken with a scan area of 100 × 100 µm. Images were processed by Gwydion 2.40.

## Data Availability

Not applicable.

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
