# Peer review of "Biological Evaluation of Photodynamic Effect Mediated by Nanoparticles with Embedded Porphyrin Photosensitizer"

_ijms, 2022, doi:10.3390/ijms23073588_

Round 1
Reviewer 1 Report
Thank you for your answers. But I still ask what was the ionic strenght of PBS? Or the composition? It can influence the stability of nanoparticles and also ROS production.
Reviewer 2 Report
The results obtained in this in vitro study showed that TPP photosensitizer encapsulated in sulfonated polystyrene NPs exhibits significant cellular cytotoxicity under PDT (blue light activation) conditions to HeLa and G361 cancer cells.
The revised paper is in good order. However, please, convert the paper to the MDPI style! In conclusion, I recommend it for publication!
Reviewer 3 Report
As the Authors neither did not number the text lines of the manuscript nor insert page numbers, it will be difficult to identify the positioning comments, for which I sincerely apologize.
The submitted paper does not follow the Journal recommendations, e.g. Authors did not use the proper Word template to prepare their manuscript.
The abstract contains more than 200 words and is not the overview of the work. A lot of acronyms present in abstract as well as in keywords section.
The title does not reflect the content of the manuscript because Authors did not prove by any technique that they “encapsulated” photosensitizer and what was the level (in percent of space) of their encapsulation. What tools / techniques were used to measure the diameter of the NP, if average diameter 15±7 nm? The spread (standard deviation?) is 50% of the value.
Introduction contains reference to published research where also no TPP-NPs manufacturing procedure can be found - see reference # 21 “Highly sulfonated polystyrene nanoparticles TPP-NPs (average diameter 15 ± 7 nm) with encapsulated hydrophobic TPP (5,10,15,20-tetraphenylporphyrin, Sigma-Aldrich) photosensitizer were prepared by a top-down nanoprecipitation method as published earlier”.
Introduction does not contain information on the research of polystyrene as a material non-toxic to cells (in general).
Conclusions are almost missing.
-----------------------------
The presented article is unfinished and uncompleted - still contains author's corrections that make it impossible to understand what is the essence of this work, e.g. “from six twelve repetitions”.
There are typos everywhere, even in the cited Authors' article ref. # 29 "Photodyagnosis"
AFM images of huge HeLa cells are a classic example of form over substance. Such objects can be visualized in any optical technique.
What is the purpose of this work? Photosensitizer (PS) dose reduction to decrease side effects? If yes, there is no comparison with a similar PS dose without encapsulation. Choosing the light exposure conditions perhaps? The results of the photodynamic interaction performed under different conditions are not shown.
There are also no studies on the toxicity of irradiated and non-irradiated polystyrene NPs (without PS) in the manuscript.
The question is what is a reference in these studies? Not NPs and not PS itself.
Why Authors used 350 LEDs instead of broadband radiation source with filter or superluminescent diodes? Due to their structure, LEDs will never ensure homogeneous distribution of radiation. Were there any additional elements, e.g. diffuser, and how did they affect the radiation intensities? An explanation that "The light source is protected by a National Patent CZ 302829 B6." is insufficient.
The use of wavelength 414 nm is extremely interesting. Usually for porphyrin ~400 nm is the Soret band, and PDT is triggered by one of the Q-bands. What is the absorption spectrum of the selected PS? How it changes after encapsulation?
In vitro experiment - whether the temperature during irradiation was checked to avoid single cells overheating?
What are the differences in the absorption spectra of PS and PS with NPs? There is no figure with these data.
How the excitation and emission spectra od PS and NP-PS look like?
No PDT irradiation source spectrum available in the manuscript or in Supplementary Materials. Please complete it with the type and manufacturer of LEDs used.
-----------------------------
A list of abbreviations and acronyms will be welcome.
References: Please include the digital object identifier (DOI) for used references if available.
Reviewer 4 Report
Dear Editor
This paper by Ludmila Žárská et al deals with the elaboration of novel polystyrene nanoparticles enriched with Encapsulated Porphyrin Photosensitizer for PDT modality treatment. In general, the paper is well written and biological results are quite interesting. However, improvement on the manuscript presentation with major revision has to be performed before to be accepted in IJMS journal.
An itemized list of comments is reported below.
- I would recommend to use an appropriate abbreviation for sulfonated polystyrene nanoparticles in view to prevent confusion with light-activated photosensitizer symbolized by “PS”
- Authors, have to include all physiochemical characterizations data (e.g., DLS, TEM, UV-Vis, FTIR, etc.) of sulfonated polystyrene nanoparticles in view to get clear argumentation of the work and results. In discussion part, I would recommend first to add and discuss all physiochemical characterizations on sulfonated polystyrene nanoparticles and then switch to biological tests.
- I was wandering, is there any advantage to use PDT in comparison to TPE-PDT? The wavelength of 414 nm is the region where biological tissue has significant absorption. What about in vivo tests in this case?
- Conclusion has to be rewrite by reminding all conclusion and major results.
Round 2
Reviewer 3 Report
Dear Authors,
If you have spectral data on the excitation and emission spectrum of the structures used, it is worth including them in supplementary materials.
The source of the UV-Vis TPP spectra (Figure S8 @ DOI: 10.1021 / acsami.9b04351) is at the high school level. Noise present in the short wave, flat line in the long wave. Interesting bands are not drawn apart and described.
Reviewer 4 Report
Dear Editor,
Despite efforts that authors provided to improve manuscript, I still have uncertainty concerning the physicochemical characteristics of NPs they used: The authors claimed “The chemical and photophysical characteristics are reported in previous publications as also mentioned in the text - material and methods - tested samples”.
First they have to mention the line (line 109). At this line they wrote “TPP-NP with encapsulated embedded TPP (3mg/ml) were prepared by modified top-down method of nanoprecipitation using sulfonated electrospun polystyrene nanofiber membranes with TPP as published earlier”. Finaly what kind of product they get NP or nanofibers. The explaination that authors provided still confused. I recommande to authors to clarify correctly this point.
Second they declared some physicochemical methods they used to describe NP (such as gravimetric analysis, etc.) (e.g., line 123). There is no interest to mention that in manuscript without data. The manuscript appears as incomplet work.
Round 3
Reviewer 4 Report
Dear Editor,
According the improvements performed by authors, manuscript can be accepted for publication.
This manuscript is a resubmission of an earlier submission. The following is a list of the peer review reports and author responses from that submission.
Round 1
Reviewer 1 Report
The results obtained in this in vitro study showed that TPP photosensitizer encapsulated in sulfonated polystyrene NPs exhibits significant cellular cytotoxicity under PDT (blue light activation) conditions to HeLa and G361 cancer cells.
The paper is well organized and structured. The results are of great interest to a large community in this field. In conclusion, I strongly recommend it for publication in its current form!
Author Response
We would like to thank you for your careful reading and positive decision.
Reviewer 2 Report
I recommend briefly describe the sample preparation.
What was the ionic strenght of PBS (what was the purity of "water" if it was qwn preparation)?
It is not necessary to explain basic chemical formula e.g. H2O2...
Conclusion: is one study sufficient to make conclusion?
Author Response
We would like to thank you for your careful reading and positive decision. Please see the attachment.
